# Black Phosphorus as Multifaceted Advanced Material Nanoplatforms for Potential Biomedical Applications

**DOI:** 10.3390/nano11010013

**Published:** 2020-12-23

**Authors:** Abhijeet Pandey, Ajinkya N. Nikam, Gasper Fernandes, Sanjay Kulkarni, Bharath Singh Padya, Ruth Prassl, Subham Das, Alex Joseph, Prashant K. Deshmukh, Pravin O. Patil, Srinivas Mutalik

**Affiliations:** 1Department of Pharmaceutics, Manipal College of Pharmaceutical Sciences, Manipal Academy of Higher Education, Manipal 576104, Karnataka, India; abhijeet.pandey@manipal.edu (A.P.); ajinkya.nikam7@gmail.com (A.N.N.); fernandesgasper16@gmail.com (G.F.); sanjay987k@gmail.com (S.K.); bharathsatavahana@gmail.com (B.S.P.); 2Gottfried Schatz Research Centre for Cell Signalling, Metabolism and Aging, Medical University of Graz, 8036 Graz, Austria; ruth.prassl@medunigraz.at; 3Department of Pharmaceutical Chemistry, Manipal College of Pharmaceutical Sciences, Manipal Academy of Higher Education, Manipal 576104, Karnataka, India; subhamdas4646@gmail.com (S.D.); alex.joseph@manipal.edu (A.J.); 4Department of Pharmaceutics, Dr. Rajendra Gode College of Pharmacy, Buldhana 443101, Maharashtra, India; pkdesh@gmail.com; 5Department of Pharmaceutical Chemistry, H R Patel Institute of Pharmaceutical Education and Research, Karwand Naka, Shirpur, Dist Dhule 425405, Maharashtra, India; rxpatilpravin@yahoo.co.in

**Keywords:** bioimaging, wound healing, 3D printing, surface modification, characterization

## Abstract

Black phosphorus is one of the emerging members of two-dimensional (2D) materials which has recently entered the biomedical field. Its anisotropic properties and infrared bandgap have enabled researchers to discover its applicability in several fields including optoelectronics, 3D printing, bioimaging, and others. Characterization techniques such as Raman spectroscopy have revealed the structural information of Black phosphorus (BP) along with its fundamental properties, such as the behavior of its photons and electrons. The present review provides an overview of synthetic approaches and properties of BP, in addition to a detailed discussion about various types of surface modifications available for overcoming the stability-related drawbacks and for imparting targeting ability to synthesized nanoplatforms. The review further gives an overview of multiple characterization techniques such as spectroscopic, thermal, optical, and electron microscopic techniques for providing an insight into its fundamental properties. These characterization techniques are not only important for the analysis of the synthesized BP but also play a vital role in assessing the doping as well as the structural integrity of BP-based nanocomposites. The potential role of BP and BP-based nanocomposites for biomedical applications specifically, in the fields of drug delivery, 3D printing, and wound dressing, have been discussed in detail to provide an insight into the multifunctional role of BP-based nanoplatforms for the management of various diseases, including cancer therapy. The review further sheds light on the role of BP-based 2D platforms such as BP nanosheets along with BP-based 0D platforms—i.e., BP quantum dots in the field of therapy and bioimaging of cancer using techniques such as photoacoustic imaging and fluorescence imaging. Although the review inculcates the multimodal therapeutic as well as imaging role of BP, there is still research going on in this field which will help in the development of BP-based theranostic platforms not only for cancer therapy, but various other diseases.

## 1. Introduction

The discovery of Black Phosphorus (BP) dates back to a hundred years ago. It all began with Bridgman [1], who brought about the conversion of white phosphorus to black phosphorus under a high temperature and pressure. Later, Hultgren et al. [2] demonstrated the orthorhombic structure of BP using X-ray diffraction and also found that BP was a thermodynamically stable allotrope of phosphorus. BP has a lot in common with graphite, including its structural, physical, and electronic properties. As graphene has a wide spread application in the biomedical field [3], similarly, BP is also being explored for its utility in biomedicine. For instance, few-layer BP (FLBP), due to its structure, photonic properties, biodegradability, and low toxicity, has been exploited for drug delivery, bioimaging, phototherapy, and combination therapeutic strategies [4]. The phosphorus atoms in BP are arranged in the form of puckered honeycomb layers connected via weak interlayer van der Waals forces and strong intralayer bonding between P atoms [5]. Its unique properties, such as tunable band gap, high carrier mobility, and anisotropic properties, has enabled its application in various physical and chemical fields [6].

The interaction between biological systems and BP has been explained to a certain level, which details basic information for modifying the physicochemical properties of black phosphorous to enhance the efficacy in therapeutic applications and reduce the unforeseen side effects. Research conducted by Chen and team [7] emphasized that by crossing the blood brain barrier (BBB) and acting as a neuroprotective agent in various diseases such as Parkinson’s and Alzheimer’s disease, BP nanocomposites can absorb excess transition metals, such as copper, by specifically protecting the neuronal cells from neurotoxicity induced by copper. Although there are many papers describing synthesis approaches and their biomedical applications for cancer, very few papers have focused on their application in different diseases. In this review, we describe the synthesis methods of BP and provide a detailed description of different surface modification strategies using peptides, drugs, polymers, aptamers, and lipids. We have also discussed several characterization techniques such as spectroscopic, thermal, optical, and electron microscopic techniques. A comprehensive section is dedicated to the biomedical applications of BP in drug delivery for different diseases such as bone therapy, wound healing, cancer therapy, and neurodegenerative diseases. The application of BP in 3D printing and bioimaging is also discussed.

## 2. Synthesis

Bulk BP is extremely stable, especially in comparison to other allotropes, due to its robust hydrophilic property, oxidation state, and its low-dimensional form—i.e., few-layer or monolayer BP is very unstable. Furthermore, heating can endorse degradation of BP [8]. In 1981, by using white phosphorus (WP) and liquid bismuth at a temperature of 300 °C below 0.5 MPa, Maruyama et al. discovered the process to synthesize BP under considerably moderate pressure [9]. Unlike the methods of synthesis using WP as described above, researchers eventually started selecting red phosphorus (RP) as a raw material for producing BP to avoid the toxicity of WP [10]. Lange et al. found, in 2007, that BP can be produced from RP by adding gold, nickel, and nickel (IV) iodide in smaller amounts at 600 °C under low-pressure state [11]. Later, in the year 2014, Kopf et al. suggested that BP can be produced with RP, tin and tin (IV) iodide additives utilizing a short-way transport reaction [12]. The Sn—SnI_4_, an RP mixture—was placed in the furnace at 650 °C. For the synthesis of BP, the mixture was cooled down to the 550 °C within 7.5 h. They succeeded in reducing the sum of side phases such as AuSn, Au_3_SnP_7_, or Au_2_P_3_, as opposed to the previous approach by Lange et al. [11].

Very recently, the fabrication of BP nanomaterials has gained a broader perspective and was divided into two general strategies—namely, the bottom-up method and the top-down method. The bottom-up approach involves direct chemical synthesis of nanomaterials using a particular precursor (red phosphorus [13]), whereas the top-down approach focuses on exfoliation of the bulk material using driving force to break it down to nanometric size by means of chemical and mechanical exfoliation (Figure 1). A well elaborated review on the synthesis of BP has been reported by Boddula et al. [14]. Hence, in the following section we have summarized the importance of different synthetic schemes, addressing drawbacks and describing the most feasible synthesis for the scale up of BP nanomaterials. In a mechanical exfoliation approach, an adhesive tape is used and monolayer BP is peeled off from bulk BP. The most commonly used methods for the preparation of BP nanomaterials along with their mechanisms, advantages, and disadvantages are described in Table 1.

The monolayers of BP synthesized from the methods such as chemical vapor deposition and mechanical exfoliation are usually situated at the edge of thick sheets and are too small to process and characterize. Along with this, the inherent problems related to phosphorous, such as weaker P-P bonds, leading to its oxidation and others, pose a challenge in developing a suitable method which is practically applicable and can be scaled up for commercial synthesis. In this context, the liquid exfoliation method serves this purpose and is widely used for the synthesis of BP and related 2D nanomaterials [15,16]. This method yields nanosheets in the form of liquid-suspension that can be further processed into nanocomposites, films, and other structures. It has been extensively used and proved to be successful in exfoliating a large variety of layered crystals such as metal oxides, graphite, and chalcogenides [16]. Liquid-phase exfoliation is the simplest method that includes sonication of layered crystals to give nanosheets using stabilizing liquids such as solvents [17]/polymer solutions [18]/aqueous surfactants [19]. The repulsive forces between adjacent polymer/surfactant-coated nanosheets provide stabilization to the nanosheets. When solvents are used for stabilization, it is presumed that solvents with either appropriate solubilities or surface energy parameters bind to nanosheets with internanosheet binding strength through van der Waals interactions [20]. This lowers the net cost of energy utilized in exfoliation. These principles of exfoliation are not specific to 2D materials and can be applied to a broad spectrum of layered compounds. In fact, this method has already been utilized for a variety of transition-metal dichalcogendes (TMDs), graphene, and molybdenum disulfide (MoS_2_) [17,21] due to a plethora of advantages such as quick and easier synthesis, being sturdy to environmental conditions, and flexibility in terms of scaling up. Several studies have already reported the biomedical applications of BP nanomaterials, such as bioimaging and phototherapy, synthesized by this method [22]. Thus, liquid exfoliation can be a potential route of synthesis for the fabrication of BP and related 2D nanomaterials at a commercial scale for biomedical applications.

## 3. Optical and Electronic Properties

Because of its crystalline lattice, the anisotropic characteristics of BP are significantly greater than most other 2D materials. Previous reports mainly focused on BP optical anisotropy in the Near infrared region (NIR) [27] until BP anisotropy was identified in the visible region [28]. A group of scientists [29] outlined a sophisticated experimental investigation on the optical anisotropy of BP by integrating existing spectral techniques and a specialized imaging technique. This exemplary tailoring could help both to determine the crystallographic alignment and characterize the surface of BP crystals. It demonstrates optical anisotropy by absorbing polarized light in the direction of the armchair, while transmitting polarized light in a zigzag path [30]. It is also observed that the optical properties are strain-dependent (amount of deformation) [31], and the optical bandgap can be tuned from one atomic layer 0.38 to 2.07 eV by the biaxial strain. Strain engineering can, therefore, regulate BP’s optical response. Furthermore, the number of stacked layers not only affects the ability of electrons to interact, but also controls the excitonic effects and optical spectra of BP. Phosphorene is used in optical linear polarizers [32] because of its ability to absorb IR and visible light and also has showed layer-dependent photoluminescence, wherein the luminescence intensity in bilayer BP was reported to be much higher than the intensity in five-layer BP [33].

Monolayer BP, on the other hand, displays excellent carrier mobility (up to 1000 cm^2^/Vs) due to its elevated sensitivity to electrical disturbance, which is superior to metal transition dichalcogenides and has a direct bandgap that can be tuned with the strain applied. This makes it possible to accurately detect incompatible and dissociated gaseous molecules, especially of toxic gases. Because of this property, BP can also be used in immunosensors in addition to gas sensors for antigen–antibody interaction detection [13]. It has already been observed that the value of BP’s bandgap increases with the reduction in the number of layers. The direct-indirect-direct transformation in the bandgap of 2D phosphorene is generated by the addition of axial pressure. For the transformation to take place in the zigzag direction, only 2% compression is required [34]. BP may also undergo semiconductor to metal transition after mild deformation [35]. In a theoretical study, it was proposed that the removal of a few atoms from the BP monolayer would result in the creation of a honeycomb-like structure consisting of two sublattices arranged in a zigzag way [36]. Such an arrangement is known as blue phosphorene which can act as an indirect semiconductor with a bandgap of ~2 eV. This tuning of the bandgap in BP allows it to have many electrical properties. Various other phosphorous polymorphs, such as δ phosphorous [37], g phosphorous [38], and β phosphorous [39], were also expected to have stabilities similar to that of BP, based on average ab initio density calculations. The incorporation of these distinct phosphorene polymorphs into a monolayer heterostructure forms the foundation for enabling dual (metallic and semiconducting) conduction in a single layer of material [40]. Therefore, phosphorene and few-layer BP could be great candidates for optoelectronic applications through atomic-level engineering [41].

## 4. Surface Modification of Black Phosphorus

As an evolving component of two-dimensional (2D) nanoparticles, BP-based nanoparticles exhibit great physicochemical characteristics and have great potential for use in modern nanomaterials. Even then, owing to their degradability and in vivo interactions with biomolecules such as plasma proteins, bare BP-based nanoparticles probably reduce their biomedical operations, majorly limiting their potential application in the biomedical arena. Surface engineering offers key techniques for the development of biomaterials of the next generation since the interactions among man-made substances and the biological environment occurs at the surface. A number of surface modifications have been established for BP-based nanoparticles using peptides, drugs, polymers, lipids, aptamers, and antibodies to overcome these drawbacks and obtain stable and effective clinical outcomes, thereby highlighting the advancement of multifunctional BP-based nanoparticles in the biomedical field for more functional use. The section below highlights the current progress in the surface modification of BP-based nanoparticles and Table 2 summarizes the various surface-modified BP nanomaterials and their advantages.

### 4.1. Modification Using Peptides

Owing to its peculiar electrical, photonic, and mechanical characteristics, 2D black phosphorus (BP) has received the attention of the scientific community over the past few years. However, BP’s inherent instability prevents its preservation and functional use. Peptides are used for molecular expression in biology and serve an important role in the development of surfaces that can recognize, react to, and eventually direct biomolecular activities at the solid–liquid interface; hence, modifying the surface of BP can overcome the abovementioned hurdles. A tailor-made tripeptide Fmoc-Lys-Lys-Phe (Fmoc-KKF) for exterior modification of BP nanosheets was synthesized in response. The BP@FKK composite exhibits exemplary stability, thereby greatly enhancing the lifespan as compared to pure BP with accelerated degradation. The BP@FKK complex also exhibited improved cellular uptake and favorable compatibility with cells due to the surface modification [42]. BP has demonstrated significant biocompatibility and immense efficiency of photothermal alteration under near-infrared light, making it quite convincing for photothermal therapy. Factual utilizations, however, are significantly hindered as BP lacks a targeting feature and readily deteriorates in cancer cells, particularly in answer to strong oxidative intracellular stress. A researcher stated that the mitochondrial targeting peptide functionalized BP nanosheets coated with an acid-labile polymer shell (doubly functional BP (DFBP) nanosheets) demonstrated exceptional stability. After surface alteration with DFBP, not only do nanosheets possess exceptional capability to accumulate in tumor tissue through surface charge switching, they could also target mitochondria, as shown in Figure 2 [43].

### 4.2. Modification Using Drugs

Black Phosphorus (BP) possesses high load carrier mobility, a configurable direct bandgap, and a distinct in-plane anisotropic framework; consequently, its applications are largely confined by the ease of BP oxidation into P_x_O_y_ species under ambient circumstances. In a previous study, modified cisplatin−Pt−NO_3_ [Pt-(NH_3_)_2_(NO_3_)_2_] was adopted for surface functionalization of BP nanosheets to produce Pt@BP (Figure 3), which preserves surface morphology and BP nanosheet properties in ambient conditions for more than 24 h. Pt@BP displays a significant cellular uptake rate and greatly improves cisplatin-resistant cancer cell lines (A2780 and HepG2) drug response relative to unaltered cisplatin [44].

### 4.3. Modification Using Polymers

The main drawbacks of BP are its insolubility in common organic solvents and poor air-stability. This problem can best be overcome by integrating BP into a polymer framework or a polymer medium for photonic and optoelectronic applications. These modifications on BP using polymers enables the development of target specific drug delivery system and increases stability and biocompatibility. Polymer-based nanoparticles are basically developed in the biomedical field to shield guest particles or molecules from deterioration and preserve their activity [50]. Polymer modulation may decrease nanoparticle’s surface energy, neutralize the surface charges, and increase the steric impedance among particles to avoid nanoparticle accumulation [51]. These properties have been shown to significantly increase the biocompatibility of BP nanomaterials including the hydrophilicity obtained from the polymeric coating [52]. Polymeric enclosures, on the other hand, can minimize the interaction of BP nanomaterials with water and oxygen and thereby efficiently reduce the degradation by oxidation [53,54]. In this segment, we discuss the various polymeric surface modification approaches on BP nanomaterials. In one study, BP quantum dots (BPQDs) were manufactured utilizing a liquid exfoliation approach which was coupled with sonic probe and bath sonication. The BPQDs displayed increased stability in the physiological medium after PEG conjugation, and no apparent toxicity was identified for different cell types [45]. In similar research, BPQDs were fabricated and functionalized by targeting moieties of folic acid linked with PEG-diamine (PEG-NH2-FA), for photodynamic-photothermal-chemotherapy by loading with anticancer drug (doxorubicin). Using PEG@BPQD@DOX, 5 min of NIR irradiation significantly raised the temperature of the tumor site to 44.2 °C since PEG-coated nontargeting NPs did not target tumor sections efficiently. Furthermore, after 4 h and an FA-PEG@BPQD@DOX intravenous (IV) injection, the temperature at the tumor section rose speedily along with the period of irradiation. The irradiation of 300 s at 808 nm (2 W/cm^2^) speedily increased the temperature to 56.8 °C, which is enough to destroy human embryonic kidney (HEK 293T) cancer cells [46]. In another approach, BP was coated with polydopamine (PDA) and a novel nanocapsule BP@PDA-PEG-FA loaded with doxorubicin (DOX) as a classical drug for cervical cancer treatment was prepared, and configured with mercapto group (HS)-based targeting polymer, HS-PEG-FA. This drug delivery system based on BP demonstrated improved stability, significantly higher photothermal performance, and the capability to kill cancer cells. In addition, DOX was released in a low pH microenvironment of tumor under NIR laser irradiation, showing the impact of synergistic therapy [47].

### 4.4. Modification Using Aptamers and Antibodies

Apart from protection, surface modifications may also enhance BP’s biosensing properties and thus lead to its probable use in biomedical applications. Aptamers show a peculiar binding to their respective target molecule based on their particular three-dimensional framework, which may be a small molecule, a macromolecule, or an entire cell. Aptamers can be utilized for surface modification of BP because of their high precision and high affinity. Poly-L-lysine (PLL)-coated BP nanosheets (BPNSs) enabled the functionalization of BP with anti-Mb aptamer (PLL-BP-Apt) that could be used as electrodes to identify cardiac biomarker myoglobin (Mb) in serum samples by means of electrochemical detection procedures, as depicted in Figure 4 [48]. In another study, electrochemical detection of circulating tumor cells (CTCs) was reported using a probe of BP@AuNPs@aptamer coupled with immuno-magnetic separation. Aptamers can precisely bind to CTCs, whereas the aptamers along with the phosphorus oxides, such as phosphate and phosphite ion (PxOy species) on BP, can react with molybdate to produce an electrochemical current, resulting in dual signal amplification. [55]. A bio-inspired structural vesicle was developed by attaching osteoblast targeting aptamers to the poly D,L-lactic-co-glycolic acid (PLGA), which further surrounded and guided the BPQDs to actively target cells. To upregulate the development of heat shock proteins and alkaline phosphatase, which synergized with deteriorated inorganic phosphates in efficient biomineralization for bone regeneration, the photothermal activity of the BPQDs was effectively explored [56]. A nominally invasive therapeutic IV catheter was developed in another study for the photothermal destruction of CTC. The catheter cavity was filled with nanosheets of BP and the layer was modified using anti-EpCAM antibody. The catheter continuously captured CTCs in the peripheral blood with the help of circulatory system and the trapped CTCs were employed downstream or eliminated in vivo by NIR-induced photothermal effect of the BPNSs [57].

### 4.5. Modification Using Lipids

Most of the inorganic materials have the limitation of a rough surface with sharp edges, which can affect cellular organelles during interactions. Similar reports have been published for graphene oxide whose sharp edges have been reported to damage blood cells. To overcome this problem, surface modification was carried out which not only helped make the surface smooth, but also enhanced the biocompatibility of inorganic materials. In the case of BP, although very scarce literature has been published for surface modification using lipids, the available literature suggests the possible enhancement in inherent properties such as fluorescence, in addition to biocompatibility. In a recently published report, it was found that lipid modifications demonstrate intense NIR-II fluorescence and could be embedded with the PEGylated cholesterol (BP@lipid-PEG nanospheres) for both in vivo and in vitro NIR-II imaging [49]. The subsequent nanospheres displayed broad emissions by an 808 nm laser under excitation from 900 to 1650 nm and had a quantum yield of 8% of that of the standard reference dye, IR-26.

## 5. Characterizations Techniques for BP

The fundamental properties of BP can be identified and exploited by several characterization techniques. For instance, UV–Vis spectroscopy can be employed to record the absorption spectra of BP at an excitation wavelength ranging from 290 to 450 nm. Raman spectroscopy aids in finding the sample purity upon irradiation by a visible laser beam with a 514 nm wavelength and also serves as an efficient method to detail the vibrational and rotational modes of the sample which in turn describes its morphology [13]. A spectrofluorometer provides the information about the internal quantum yield by excitation and emission wavelengths of 370 nm and 380–700 nm, respectively. Scanning transmission electron microscopy confirms the exfoliated structure of BP by resolving the atomic structure with a precise resolution of 2 Å. Hence, in the following section, we describe a few techniques employed in characterizing the BP nanosheets in detail.

### 5.1. Spectroscopic Techniques

Several spectroscopic techniques, such as infrared spectroscopy, X-ray photon spectroscopy, polarization-dependent spectroscopy, and photoluminescence (PL), are employed in characterizing BP. These techniques help in the understanding of representative vibrational modes, crystallinity, crystallographic directions, and intrinsic bonding energies. External stimuli such as temperature, strain and pressure are used to explore the inherent anisotropic properties which arise because of the specific puckered structure of black phosphorous [58]. The spatial inhomogeneity of phosphorene is of specific interest because of its puckered structure; in addition, the strain can alter properties such as mechanical and chemical properties. Near field Raman or confocal Raman scattering offers higher resolution in the spatial strain distribution. Photoluminescence (PL) often depends on the BP energy band. Semiconductors can adsorb photons with adequate energy that leads to several pairs of electron-holes on the surface of materials—this is called intrinsic absorption. Such extra carriers will disperse and recombine as soon as possible into the inner region of the semiconductors. This generates photons in the recombination [59].

PL calculation is, therefore, an important method for measuring the structure of the band. The PL peak position differs according to BP thickness. When the exciting laser’s polarization and the polarizer’s linear polarization are parallel and align with the orientation of the armchair (x-axis), the photoluminescence intensity will reach its maximum. In addition, the PL strength is strongly reliant upon the extinction ratio [60]. If the polarization of the detector is in a zigzag path, the intensity drops significantly—that is, less than 3% in the direction of the armchair, irrespective of the exciting light’s polarization. These unusual anisotropic characteristics distinguishes the BP dramatically from other 2D materials and can shed some light on the dynamics of the carrier and the band structure [60]. The PL spectroscopy of BP was performed by Zhang and team on a silicon substrate. Stronger PL peaks were observed at 1558, 1413, 1268, and 961 nm, in 5-, 4-, 3- and 2-layered flakes, which correspond to the energy peaks of 0.8, 0.88, 0.98, and 1.29 eV, respectively [33]. These PL peaks demonstrated the nature of extinctions, representing the lower bond on the primary band gap values present in few-layer BP. The position of energy in the measured peaks increases with the decrease in number of layers and indirectly denotes that, with the decrease in layer number, the band gap of few-layer BP increases, which may be attributed to the smaller band dispersions of the conduction band and valence band related to the weaker interlayer interaction [61]. Another study reported the PL of BP upon excitation with a green laser and the results showed that with increase in number of layers the intensity of the PL peak decreased, indicating a decrease in optical bandgap [62].

### 5.2. Thermal Techniques

Understanding the thermal properties of few-layer or single-layered BP is of particular interest for fundamental research and its applicability. From the research perspective, it is important to know the conduction of heat in low-dimensional materials to obtain a better picture of thermal management and heat dissipation at the nanoscale [63]. Phonons (vibrations of crystal lattice) function as the primary heat carriers in insulators and semiconductors. This phenomenon of phonon conduction in nanowires and films differs from the bulk and is suppressed because of the changes in phonon density, dispersion, and boundary scattering. So, the possibility of controlling the phonon at the nanoscale has led to the application of nanomaterials in thermoelectrics [64]. Although the melting point (M.P.) of bulk black phosphorous ranges between 600 and 1000 °C [65], the precise M.P. for a few-layer BP remains under consideration. The information of BP’s linear thermal expansion coefficients (LTECs) seems to be of great importance in terms of 2D materials, to acquire a better and deeper understanding of its anisotropic thermal properties. Presently, there are very few techniques to measure the thermal conductivity of BP. Temperature-dependent Raman scattering was performed by a research team [66] and the results showed that the temperature coefficient of BP thin films decreased with decreasing thickness. In addition, the suspended state BP showed thermal conductivity of 15.8 W/mK and it increased to 29.2 W/mK when supported on silica substrate. This indicates that the properties of BP films can be significantly affected by the substrate and the extent of this varies based on the property of the specific material of a given thickness. Jeon et al. experimentally and theoretically explored thickness influences on the thermal conductivities of BP nanosheets for an extensive temperature range of 100 to 300 K [67]. Henry et al. recently reported precise measurements of thermal expansion along the three crystallographic axes, using X-ray diffraction in situ at high temperatures [68]. Another group of researchers utilized micro-Raman spectroscopy to study the thermal conductivity of BP films and found that the anisotropic ratio for thick films was ~2 and dropped down to ~1.5 for the thinnest film, which had a thickness of 9.5 nm. This can be ascribed to the anisotropic phonon dispersion. The advantage of using this method is that it allows for simultaneous probing to record temperature as well as for local heating of the thin film sample. However, difficult automation and uncertainty of beam size are limitations [69,70]. Hence, there is a need for reliable experimentation and systematic comparison between the measurements and theoretical predictions to understand the underlying physics in a better way.

### 5.3. Optical Techniques

2D materials show a good optical response in a third order nonlinearity with a broad bandwidth, a miniature size, and an ultrafast response. Hence, they are suitable for optical light processing operations [71]. The same as photon radiation, the intrinsic essence of optical absorption includes an interaction between an electron and a photon usually occurring at edges of the band. As stated by Fermi’s golden rule, this process has a higher probability of occurring in two energy states having a higher combined density of states [72]. Optical properties (OPs) rely primarily on different factors, such as Raman scattering (RS), optical absorption, and BP photoluminescence [60]. BP’s optical absorption has been extensively researched both by theoretical and experimental measurements [30,60,73]. BP’s direct bandgap lies between graphene and transition metal dichalcogenides, and thus, it serves as an ideal material for several optical applications. The BP bandgap depends heavily on the number of layers. Therefore, the size of the respective photoabsorption often varies according to the thickness of BP flakes.

Raman spectroscopy is a robust, nondestructive tool often used for characterizing the nanoparticles to determine the thickness [74], strain [75], and angle [76] of the nanomaterials. Based on the group theory and conservation of momentum, BP consists of six Raman active modes. However, when the incident laser falls perpendicularly to the layers of phosphorene, only three modes can be identified, which are A^1^_g_, A^2^_g_, and B_2g_ phonon modes corresponding to the 362, 467, and 439 cm^−1^ wavelengths, respectively [62]. When the temperature rises from 78 to 573 K, softening occurs in these modes [77]. With the increase in number of layers, Raman peaks red-shifted, indicating the hinderance of phonon vibration with the addition of new layer to the surface of BP [78]. Further, based on optical anisotropy, the crystalline orientation of BP can be identified using polarized (angle resolved) Raman spectroscopy [76]. It was observed that by rotating an arbitrarily located sample under the parallel polarization configuration, the intensity of the A^2^g band increased when the orientation of armchair was along the polarization path of scattered light. Anisotropic Raman spectroscopy also finds an application in measuring the degree of strain on BP samples [71,75]. The Raman scattering of BP displays that the thickness of BP layers can be identified by the low-frequency interlayer Raman modes. As described by the optical transition selection rule, each interaction can have different anisotropies. Thus, using the multiple characterization techniques, the optical characterization provides an efficient platform to understand the physics of materials.

### 5.4. Electron Microscopic Techniques

Electron microscopy offers enhanced magnification and resolution for the visualization of biological and nonbiological samples and understanding their structural organization. Transmission electron microscopes (TEMs) function by transmitting the beam of electrons through the sample generating a projected image, while the scanning electron microscope (SEM) scans the surface of the sample with the aid of a focused electron beam, which renders the emission of particles from the surface forming an image [79,80]. Generally, electrons with short wavelengths are used as a source of illuminating radiation. The same as other nanoplatforms, BP can also be surface characterized using these electron microscopic techniques. It shows a puckered hexagonal ring in the orthorhombic crystal structure and, as described earlier, the individual layers are held together by van der Waals forces. TEM images of a single layer of BP are shown in Figure 5 [81]. For instance, BP quantum dots (BPQDs), when analyzed through TEM, demonstrated the lattice fringes of 0.34 nm attributable to the (021) plane of the BP and the statistical analysis of the obtained TEM analysis revealed that the average size and thicknesses of the BPQDs were 2.6 ± 1.8 nm and 1.5 ± 0.6 nm, respectively [45]. Another study employed SEM to analyze the cell viability of osteo-sarcoma derived (SAOS-2) cells and healthy osteoblast cells (HObs) upon incubation with the exfoliated BP. The initial morphologies of the SAOS-2 cells were replaced by the shape of typical apoptotic cells, while the images of HOb cells exhibited polygonal morphologies and good cell spreading of the differentiating cells [82]. Similarly, FLBP nanosheets fabricated using the liquid exfoliation method were subjected to TEM analysis by drop-casting the exfoliated FLBP suspension on a carbon grid. TEM images showed that the nanosheets possessed an average lateral length of 2.1 mm and thickness of around 8nm [83].

### 5.5. Plasma-Protein Adsorption Study

The adsorption of proteins on the surface of nanoparticles when they come in contact with the protein rich environment plays a vital role in deciding the fate of NPs. Several studies have described the role of protein adsorption in biological interactions [84] and have stated that “Protein corona” refers to the spontaneous self-assembly or biomolecular coating of the proteins on the surface of nanoparticles [85]. The formation of corona not only alters the physical properties of the nanomaterials such as size, surface charge, shape but also alerts the immune system of the body to recognize the nanoparticles as an intruder and hastens the process of opsonization [86]. Having said that, it becomes important to identify various characterization techniques to detect the extent of protein corona formation. Though several techniques such as quantitative and qualitative proteomic assays have been described to characterize the protein corona formation of the nanoparticles [87], very few studies have reported on its formation around BP nanomaterials. Here, we have discussed the techniques available to characterize the formation of protein corona around the BP nanomaterials.

Electron microscopies, such as TEM, provide a clear picture of protein corona formation around the nanoparticles. For instance, in a previous study [88], the size of BPNSs was increased by 11.9 nm after being covered by plasma proteins and, in another study, the size increased by 30 nm [89]. The corona formation can also be detected by a change in surface charge. The surfaces of BPNSs coated with plasma proteins were entirely different from the native form and the charge changed to a positive value from negative. Liquid chromatography with a mass spectrometry detector was used to qualitatively analyze the components of the BP–corona complex. The results revealed that the binding of proteins to BP directly depends on the protein size and, additionally, the BPNSs mainly bind with immune relevant proteins. The composition of the protein corona can also be evaluated using Sodium dodecyl sulfate-polyacrylamide gel electrophoresis (SDS-PAGE). The strength of the bands in electrophoresis was examined by relative densitometry and the results demonstrated that the intensity of bands increased with the increase in protein concentration [90]. The outcomes of dynamic light scattering studies showed that protein corona formation increased the size of BPNSs by approximately 30 nm and the ultrasmall BPQDs were redefined to bulky spherical QDs after protein corona formation.

Elemental dispersive spectroscopy can also serve as a prominent method to characterize the formation of a protein coat around BP nanomaterials. The increase in the intensity of C, P, and N indicate the formation of corona [88,89]. Further, a real-time polymerase chain reaction can be used to understand if the BP nanomaterials induce the M1 macrophages. When BPNSs were exposed to the macrophages, there was an upregulation in the M1-related markers such as interleukins and CD16. With the enrichment of immune proteins, the uptake of the BP–corona complex by macrophages increases, and this, in turn, enhances the immune perturbation in the macrophages, while the expression of M2-related genes, such as IL-10 and CD206, Z decreased. These data help the researchers to understand the process of protein corona formation and its role in deciding the biological activity of BP nanomaterials and thus enable them to design safe and effective BP nanoplatforms.

## 6. Biomedical Applications of Black Phosphorus

### 6.1. Application in Drug Delivery

BP, also known as phosphorene, has drawn significant interest in therapeutics and drug delivery amongst the 2D class of inorganic nanomaterials [91]. The puckered orthorhombic configuration provides BP with a high surface to volume ratio, thus increasing the drug loading capacity [92]. Because of phosphoric acid, the surface of BP is negatively charged and can thus encapsulate positively charged drug molecules or nanoparticles by electrostatic interactions within the interlayer spaces [93,94]. The use of BP in drug delivery varies from bone therapy (owing to the inclusion of a phosphorus atom that is an integral part of the bone) [95], cancer therapy (high surface area and photosensitive characteristics) [96], wound healing (superior photothermal agent and oxygen carriers) [97], neurodegenerative disease (antioxidants and enhanced BBB permeability owing to BP’s photothermal properties) [7], and implants [98]. The following section describes the application of BP in drug delivery.

#### 6.1.1. Bone Therapy

Bone regeneration is a significant challenge in clinical surgery due to various conditions such as osteomyelitis, osteitis, traumas, and tumors [99]. In the past, several therapeutic strategies (autographs and allografts) have been implemented, with autographs being the gold standard of clinical practice. However, this suffers from certain disadvantages such as damage to the donor site, reduced bone mass and size, low availability, and the fact that it can induce immune responses and transmit disease [100]. Hence, with the advancements in tissue engineering, synthetic bone replacement materials have become the primary study in the field of bone regeneration [101]. On comparison with autografts, synthetic bone materials pose several advantages such as bone targeting, drug delivery carriers [102], ease of synthesis [103], and ease of functional modifications [104].

Phosphorus (P) is a key element of the body responsible for bone regeneration and for maintaining a bone’s mechanical strength [105]. It accounts for approx. 1% of the overall body mass [106], 85% of which is found in the form of hydroxyapatite in the bones and teeth [107]. Numerous studies have shown that materials rich in phosphate can promote mineralization and promote bone regeneration [108]. BP-based nanomaterials have substantial benefits in bone therapy due to the single P element that generates nontoxic PO_4_^3−^ when degraded in ambient conditions [109], which is a source for remineralization that can promote in situ bone regeneration by trapping the surrounding Ca^2+^ ions by forming calcium phosphate [110]. Another advantage is that the photosensitive nature of BP in NIR allows the release of entrapped drugs in a controlled manner for a longer duration. Moreover, few studies have shown that BP’s photothermal property tends to upregulate markers—i.e., alkaline phosphatase [45] and heat shock protein [111]—that are responsible for accelerating bone growth. BP 2D nanosheet scaffolds, hydrogels, and microspheres have been employed in bone regeneration. In a study, porous Gelatin methacryloyl (GelMA) hydrogel-based BP nanosheets (BPNS) (previously synthesized by liquid exfoliation) were fabricated to promote the release of phosphate ions in a controlled manner and trap the surrounding Ca^2+^ ions to facilitate bone regeneration. In vitro and in vivo experiments (rabbit model) were performed to recognize the photoresponsive phosphate release activity and to determine the efficiency of bone regeneration. The findings showed that BPNS hydrogel release P in a photoresponsive and controlled manner, speeding the process of in vitro mineralization. In addition, the in vitro osteogenic differentiation results also showed that the BPNS hydrogel improved osteogenic differentiation of stem cells via the BMP-RUNX2 pathway. The in vivo studies revealed that when the BPNS hydrogel was used, bone recovery was possible within 12 weeks [95]. Similarly, Miao et al. synthesized BPNS via a liquid-phase stripping method and integrated it onto GelMA hydrogel to form BP/Gel nanocomposites and explored its activity on bone regeneration. The in vitro experiments have shown that incorporating BPNS into the hydrogel matrix promotes biomineralization, which was confirmed by quantifying P, O, and Ca elements by SEM equipped with energy dispersive X-ray spectroscopy. Furthermore, staining with Alizarin red S staining verified that the nanocomposite could facilitate osteogenic differentiation of hMSCs. To study the in vivo bone regeneration using BPNS hydrogel composites, a rat cranium defect model was used. Bone growth was being observed to advance from the periphery to the center in 9 weeks, thus illustrating the in situ development of bone [112]. In a study, a thermo-responsive hydrogel was constructed by incorporating BPNS into a chitosan polymer with platelet rich plasma for phototherapy of rheumatoid arthritis (RA). The Photodynamic Therapy (PDT) and Photothermal Therapy (PTT) properties of BPNS could excise the hyperplastic synovial tissue and eventually promote in situ biomineralization. In addition, a delayed drug release of methotrexate was also detected, indicating synergistic effects in the treatment of RA and stimulation of bone growth [54].

Matrix vesicles (MVs) derived from extracellular vesicles, which are primarily responsible for building the extracellular matrix and controlling the physiological activities of cells, were responsible for bone mineralization. In a study, similar concept was used where, aptamer-modified MVs were designed for in vivo bone regeneration and biodegradable BPQDs served as a source for inorganic phosphate and MVs to direct the complex to the desired region in the bone [56]. In a previous study, BP and strontium chloride were embedded into poly D,L-lactic-co-glycolic acid (PLGA) microspheres that were synthesized by an o/w emulsion solvent evaporation technique for bone regeneration, as shown in Figure 6. The excellent PTT effect of BP assisted NIR irradiation led to the controlled release of Sr^2+^ by penetrating the PLGA shell. In vivo results on the rat femoral defect model by implanting the BP microspheres showed enhanced bone regeneration compared to plain microspheres or without NIR irradiation [113]. In bone regeneration therapy, BP-based nanomaterials have demonstrated satisfactory outcomes relative to other 2D nanomaterials. However, some problems, such as low stability and unpredictable immune responses, are yet to be addressed.

#### 6.1.2. Cancer Therapy

The novel application of 2D metallic nanomaterials as nanocarriers in multimodal nanomedicine is being thoroughly investigated as an alternative to conventional cancer therapy due to their exceptional physicochemical properties. BP has received significant research interest among the different types of 2D metallic nanomaterials due to its unique characteristics [114]. For example, BP metallic nanocomposites offer great potential as photothermal agents for PTT (due to the high photothermal conversion) [115] and PDT (good photosensitizers due to generation of reactive oxygen species) [92,116], drug delivery (because of its high surface area, biocompatibility, and biodegradability) [117], and multimodal theranostic agent in cancer therapy [118].

NIR irradiation-mediated hyperthermia due to the presence of photothermal absorbing agents is the mechanism in photothermal therapy to eradicate tumors [119]. The photothermal conversion efficiency and high extinction coefficient of BPs make them better candidates in photothermal therapy [120]. In a study, a sprayable BPNS incorporated within a thermosensitive and biodegradable hydrogel was utilized for postsurgical treatment of cancer. The BPNs were fabricated by a modified liquid exfoliation method and further incorporated in the Poly(d,l-lactide)-poly(ethylene glycol)-poly(d,l-lactide (PLEL) hydrogel by ultrasonication in 30% *w/v* PLEL solution. On application of a small amount of hydrogel on the tumor site, rapid conversion of sol-gel occurs with NIR irradiation (808 nm) and the high PTT efficacy helps to efficiently eradicate tumor tissues, which was evident in the in vivo and in vitro studies. The biocompatible nature of the BP@PLEL hydrogel was evident upon incubation with several cancer cell lines which demonstrated that cytotoxicity and similar results were observed in vivo; hence, they concluded that the nanocomposite not only eliminates the tumor, but also assists in wound healing and has significant therapeutic potential in cancer therapy [121]. To prevent degradation of BPNS in ambient conditions, Li et al. modified its surface with 1-pyrenylbutyric acid and conjugated with RGD peptide for targeted PTT therapy (Figure 7). The RP-p-BPNS was fabricated by a simple liquid exfoliation method and the fabricated RP-p-BPNS exhibited excellent biocompatibility and good stability in an ambient environment (7 days) with carboxyl groups available at its surface for conjugation with an RGD peptide (EDC-NHS chemistry) for enhanced tumor targeting. P-BPNS showed a significant cytotoxicity effect on the cancer cell under NIR irradiation at 808nm due to the specific binding of the RGD peptide and α_v_β_3_ ligands present on the surface of the cell membrane, thereby increasing the intracellular concentration of p-BPNS in the cancer cells. Similar findings were obtained in in vivo studies of tumor-bearing nude mice in which complete tumor suppression was observed in the p-BPN-treated group in comparison with the controlled group, suggesting that the PTT effect of BPNS was further amplified by RGD conjugation [122]. In a recent publication, BP’s PTT potential was explored by synthesizing BPNS gold nanoparticles (Au-BPNS) using a simple one-step synthesis. The experimental findings showed that, upon irradiation with 808 nm laser, Au-BPNS showed an excellent antitumor effect in the NIR region compared with bare BP. Studies have also shown that Au-BPNS could be utilized as an efficient surface-enhanced Raman scattering (SERS) matrix for Raman biodetection [123].

In addition to BP phototherapy, chemotherapy was found to be effective wherein BP appeared to have chemotherapeutic effects and could also be loaded with several molecules for enhanced chemotherapeutic activity [124]. Several papers report the use of combinatorial therapy that includes PTT/chemotherapy, PTT/gene/chemotherapy, and PTT/PDT/chemotherapy, as shown in Table 3.

Using a modified liquid exfoliation technique, Wan and colleagues developed BPNS and coated it with PEG through electrostatic deposition. In order to boost immunotherapy, BPNS was integrated with R837. In the presence of NIR (808 nm), the developed PEG-BPNS displayed excellent photothermal stability and high photothermal efficiency, thereby inhibiting B16 cells (murine melanoma cell lines). Compared to bare BP, the synergistic effect of PTT generated tumor antigens and R837-mediated activation of antigen-presenting cells caused a significantly greater response [125]. The high ROS generation capability of BPNS makes them effective photosensitizer agents and ideal candidates in PDT. In a previous study, a novel nanocomposite (BP@Au@Fe_3_O_4_) was fabricated by depositing Au and Fe_2_O_3_ nanoparticles on BPNS using a single-step electrostatic attraction method. The BPNs were synthesized using a liquid exfoliation in deionized water for 8h. The BP@Au@Fe_3_O_4_ showed good antitumor activity due to synergistic activity displayed by both Au (PTT and PDT at 650 nm) and Fe_2_O_3_ (tumor targeting and magnetic resonance imaging). The MTT assay was used to confirm the noncytotoxic effect of the prepared nanocomposite in HeLa cells thereby exhibiting good biocompatibility. The in vivo results proved the antitumor efficiency of the nanocomposites [126]. Hence, several studies proved BP efficacy as a PTT, PDT, and chemotherapeutic agent in cancer therapy [53,126,127,128,129,130,131].

#### 6.1.3. Wound Healing

The complicated wound healing process begins immediately after the injury and regains the integrity of the skin within 12 weeks. Inflammation, new tissue growth, tissue remodeling, and regeneration are the three phases in the wound healing process. [140,141]. An impairment in the wound healing process can cause serious complications and lead to disease. [142]. With the increase in side effects that mainly affect the liver, the use of conventional pharmaceutical drugs for a longer period has been a major concern in the medical field. [143]. As a result, the need for an accelerated wound healing process began with the use of natural polymers such as polysaccharides and proteins but lacked bioactivity and the potential to enhance the wound healing process. Thus, extensive research has been carried out on surface-architected nanomaterials integrated in the polymeric network to obtain nanomaterials that augment the wound healing process [144]. A study reports the use of BPNS exfoliated using silk fibroin (BP@SF) as a novel wound dressing for antibacterial and rapid wound healing via NIR-mediated photothermal therapy. The BPNSs were synthesized using liquid exfoliation using silk fibroin as the exfoliating agent. The cytotoxicity assay was tested using human synovial fibroblast cells while the antimicrobial activity was assessed using *E. coli* and *B. subtilis* as representative models. In vivo studies were carried out on Kunming mice by making a 5 mm^2^ wound and infecting it with an *E. coli* suspension. The SF molecules were found to be effectively bonded to BPNS by strong hydrophobic interactions with particle sizes of 200 nm. The BP@SF were found to be stable for 14 days unlike the BPNS synthesized using N-methyl-2-pyrrolidone (NMP) which degraded within 7 days on exposure to air and water. In vitro PTT displayed good antimicrobial activity of BP@SF compared to BP@SF without irradiation, while in vivo results showed remarkable wound repair and regeneration with complete healed wound within 5 days [145]. Since oxygen is a prerequisite for quick wound healing, research findings have shown that several oxygen carriers can supply O_2_ to the surface wound that does not reach the internal layers of the wound and also find difficulty in controllable delivery of O_2_, thereby affecting its practical performance. Therefore, Zhang et al. prepared microneedles loaded with BPQD and Hb for the NIR-mediated controllable delivery of oxygen to the skin to promote wound healing as shown in Figure 8. BPQDs were loaded onto the microneedle tips which were constructed of GelMA and PVA as the backing layer. The in vitro studies were carried out using a standard fibroblast cell line while the in vivo studies were carried out on Type 1 diabetic rat infected with a 1cm thick cutaneous wound. On application of the microneedle to the affected skin, the PVA backing layer dissolved within a few minutes, causing the GelMA tips to be entrapped in the wound. On exposure to NIR, the BPQDs increased the local temperature to 50 °C within 2 min, causing the Hb to release oxygen in a controllable manner. The in vivo results showed complete wound recovery within 9 days, demonstrating the potential for BP in wound associated pathological events [97].

#### 6.1.4. Neurodegenerative Disease

Neurodegenerative conditions are a group of uncurable heterogeneric disorders, leading to a gradual deterioration of nerve cells, affecting the central nervous system and the periphery nervous system. Tauopathies, amyloidosis, synucleinopathies, and Transactivation response DNA binding protein (43 TDP-43) proteinopathies are the most widely mentioned neurodegenerative disorders [146]. Studies have also reported that depressive symptoms are commonly observed in patients with cognitive disorders such as Alzheimer’s and Parkinson’s disease [147,148]. The treatment of such pathological brain circumstances is one of the most complicated aspects of modern medicine due to failure of maximum therapeutic compounds to effectively cross the BBB [149,150]. An ideal drug delivery carrier system would accumulate within the brain thereby crossing the BBB and releasing the drug to the specific targeted cells [151]. Recent studies have explored the potential of BP-based nanomaterials to improve the permeability of BBB to NIR exposure via PTT, since it can be used as an effective platform for drug molecules to be delivered intracerebrally. In a previous study, Paeoniflorin (Pae), an antiparkinsonian drug, was loaded within BPNS conjugated to Lactoferrin (brain targeting ligand) for systemic delivery (Lf@BP-Pae), as shown in Figure 9. Using liquid-phase exfoliation, BPNSs were synthesized and the complex had a particle size ranging between 70 and 250 nm. The nanocomplex exhibited superior PTT efficacy with a temperature elevation of 41 °C within 480 s of irradiation. The in vitro neuroprotective efficacy was tested on human neuroblastoma SH-SY5Y cells, which revealed no in vitro toxicity while the complex was primarily localized in the mitochondria. In in vitro and in vivo experimentations, the increased BBB permeation of the nanocomplex and drug release on NIR irradiation at 808 nm was clearly evident. The mechanism of BBB permeation was found to be endocytosis followed by transcytosis. Additionally, the complex prevented neuronal damage, thereby portraying an antioxidant mechanism. These findings demonstrate that the fabricated complex can be utilized for systemic delivery of antiparkinsonian drugs or drugs for the therapy of other neurological diseases [152].

Jin and colleagues developed a BPNS-based drug delivery platform for synergistic PTT/chemotherapy of depression using fluoxetine as the antidepressant drug. The BPNSs were synthesized using a liquid exfoliation technique. The BPNS had a particle size of 200 nm proved by SEM and TEM analysis. It was observed that, under 808 nm NIR irradiation, the temperature of the BPNS solution raised with dose and approx. 90% of the drug was released within 30 min of irradiation. The BPNSs were highly biocompatible and showed no toxicity towards HUVEC, LLC, 4T1, and U251 cells. The in vivo results showed that BPNS loaded with the drug shortened the therapy duration compared to the conventional fluoxetine treatment [153]. Recent studies have shown that increased levels of biometals (transition-metal dyshomeostasis), especially copper, have led to the onset and progression of neurodegenerative diseases [154]. In a recent study, BPs were utilized as Cu^2+^ scavengers for neurogenerative disorder therapy. BPNS enhanced BBB permeability via PTT irradiation, therefore increasing the drug efficacy. In the physiological state, BPNS was found to serve as an antioxidant to minimize the accumulation of Cu^2+^ dyshomeostasis-related cytotoxic ROS. BPNS also possesses high stability and excellent biocompatibility, bestowing its biosafety in neurodegenerative disorders [7].

### 6.2. BP and 3D Printing

The 3D printing industry, considered as a technological revolution, offers great opportunities in a variety of arenas including tissue engineering. Various biomaterials, including artificial scaffolding, have been widely built up for bone regeneration by applying advanced techniques for design and production [155]. Nevertheless, studies related to the application and design of 3D printing technology for targeted therapies of cancer are still in infancy. So far, there are hand countable research papers available addressing the nano biointeractions of phosphorene with body fluids as a specific biotrigger for osseointegration and osteogenesis. The concept behind such an interaction is to transform the degraded BP nanosheets into P-agents (in situ) capable of improving bone regeneration. In this context, a novel therapeutic system was designed to formulate a bifunctional BP-Bioglass (BG) scaffold by integrating the BP nanosheets in a 3D-printed BG, for the phototherapy of osteosarcoma followed by bone regeneration. First, following the implantation of a BP-BG scaffold into the bone defects resulting from surgical resection, the photothermal treatment was implemented out on the basis of the exceptional photothermal efficiency of BP nanosheets. Subsequently, BG 3D-printed scaffolds exhibit their vital roles in the successful accomplishment of cell proliferation, vascularization, angiogenesis [155,156], and differentiation, owing to their intrinsic properties for osteoconduction, osteoinduction, and osteogenesis [157]. Eventually, the scaffolds slowly degrade as new bone tissue materials, followed by the reconstruction of pathologically changed areas. The bifunctional BG scaffold supports both BP nanosheets and tumor-induced flaws in the bone. The inherent physicochemical characteristics of BP nanosheets offer an exceptional in situ biomineralization ability to aid the process of osseointegration, and the phosphate ions released in the process of degradation of BP act as anionic ligands to interact with calcium ions and promote the formation of calcium phosphate nanoparticles that assist in bone regeneration and increase biomineralization [158]. Similar studies were performed in another studies, wherein 3D-printed BPNS-BG scaffolds were developed for the photothermal therapy of osteosarcoma and bone regeneration. On NIR irradiation, the BP scaffold caused degradation to phosphate via PTT and bound the surrounding calcium to achieve biological mineralization. The in vitro and in vivo results proved that a BP scaffold has high osteogenic potential and plays critical roles in cell proliferation, differentiation and vascularization, thus it can be used for dual purposes—i.e., to treat osteosarcoma and act as a matrix to support BPNS [158].

The synergistic effect of BP and graphene oxide nanosheets on cell osteogenesis was assessed in a previous study. BP was initially wrapped in the nanosheets of negatively charged GO and then adsorbed together onto three-dimensional scaffolds of poly(propylene fumarate). GO nanosheets offer an increased surface area that would help in enhancing the cell attachment. Moreover, a continuous release of phosphate resulting from slow oxidation of the BP nanosheets packed in GO layers functions as a major facilitator of osteoblast differentiation to trigger cell osteogenesis towards a new bone formation. Results of various studies performed in the study proved that synergistic combination stimulated the osteogenesis and cell proliferation, rendering the method a promising strategy for tissue engineering applications [159]. In another research, a multifunctional scaffold was developed based on the strategy of “first kill and then regenerate”. To achieve sequential elimination of the tumors with the suppression of recurrence and enhanced tissue regeneration, the w/o polyester emulsion inks incorporated with various functional agents, was developed. The tumor resection-induced defects in the tissue were addressed by using BP nanosheets (photothermal agent), doxorubicin (anticancer agent), beta-tricalcium phosphate (proliferation and maturation of the mesenchymal stem cells derived from the bone marrow), and osteogenic peptide (osteogenic factor). The developed scaffolds had a porous biomimetic assembly, photothermal effect, and adequate mechanical strength, and showed a controlled release of peptide and DOX. The use of such a multifunctional scaffold in the tumor-bearing nude mice resulted in the elimination of tumors and prevented its recurrence. Furthermore, the osteogenic differentiation of mesenchymal stem cells derived from rat was achieved, owing to the synergistic effects of sustained peptide release and the bony environment, and BP nanosheets were found to reduce the long-term toxicity associated with the release of DOX during bone regeneration [160].

### 6.3. Bioimaging

Photoacoustic (PA) imaging provides superior contrast images and sensitivity, high-dimensional resolution, and 3D imaging which are depth-resolved. Thus, it functions as one of the emerging noninvasive cancer imaging techniques. Consequently, it has been demonstrated to be exceptional in many other conventional optical imaging techniques that help imaging-guided therapy [161]. PA is capable of generating optical contrast images with high-dimensional resolutions in tissue areas with a depth of 5–6 cm and manages background-less tumor recognition [162]. Since the PA signal initiates from the tumor location, the signal amplitude, especially at the initial stage of cancer, is considerably low. Therefore, efficient exogenous contrast agents have been researched for in vivo PA tumor imaging, such as NIR absorbing nanomaterials [163,164]. Because of its remarkable electronic and optical characteristics, BP have been recognized as an effective substance for bioimaging. The tunable band range for this substance varies between 0.3 and 2.0 eV, representing the values from bulk to a monolayer, respectively [165]. A multilayer BP may therefore enable photodetection over a large insubstantial area [166]. As outlined above, BP is very responsive to oxygen and water surface alterations can be carried out to upgrade its stability. One study, for instance, demonstrates the surface alignment of a sulfonic ester form of titanium ligand (TiL4) along with BPQDs to boost its stability for utilization as PA agent for cancer bioimaging [167]. TiL4-synchronized BPQDs demonstrated greater stability in a water dispersion relative to the plain BPQDs. Because of their strong NIR extinction coefficient, they even demonstrated better PA efficiency than AuNPs. In particular, BPQDs can accumulate gradually at a large concentration in the tumor through increased permeability and retention (EPR) effects. The high spatial resolution and exceptional sensitivity towards tumor detection illustrate their potential utilization as contrast agents for PA imaging in MCF-7 tumor cells and their applicability as photodetectors for hyperspectral imaging in biomedical studies [167,168]. The utilization of polyethylene glycol-(PEG)ylated BPNP for both PA imaging and cancer treatment applications was previously recorded in a study [166]. The surface coating of PEG and the fractional oxidation of BPNPs allow exceptional water solubility and stability. In vivo study displayed that after 24 h postintravenous injection of PEGylated BPNP, the tumor reserved a higher signal intensity in comparison with kidney and liver. Consequently, by the impact of enhanced permeability and retention, the nanomaterials have a long retention span in the tumor, whereas they can also be gradually extracted from the kidney and liver. Such characteristics make them vastly appropriate for cancer PA imaging.

## 7. Biodegradation and Toxicity of BP

BP has been extensively used in field-effect transistors and solar cells, and this has ignited a surging wave of interest in the exploration of BP in biomedicine especially as imaging agents, photothermal agents, cancer theranostic agents, drug carriers, and in bone regeneration [78]. However, in the physiological environment, the biological effects of BP-related nanomaterials are not well established, leading to toxicity and biosafety concerns, which limits their clinical applications [169,170]. Previous research claims that the potential toxicity of BP nanomaterials in biological systems to be size- [171] and layer-dependent [171,172,173]. In this context, the biocompatibility of BPQDs was assessed in a study in which the authors used HeLa cells to conduct BP nanodot cytotoxicity at concentrations ranging from 0–3 mg/mL via MTT assay. The findings revealed that the BP nanodots exhibited little or no in vitro cytotoxicity at 1.0 mg/mL; however, at 3.0 mg/mL, cell necrosis was observed due to the crystallinity of BP dots, implicating good biocompatibility of BP nanodots at lower concentrations [174]. In contrast to graphene, few research trials have demonstrated the lower toxic potential of BP as it quickly degrades to phosphorous ions in the presence of water and oxygen, which also acts as an added aid in the case of bone regeneration [175]. Another study revealed the cytotoxicity of PEG-modified BPQDs to different types of cells such as 293T, MCF-7, and C6 (glioma) cells. The findings revealed that the BPQDs did not exhibit cytotoxicity even at a concentration of 200 ppm on either of the cell lines, indicating the suitability of BP for biomedical use [45]. To address the challenges associated with toxicity, the surface functionalization of BP has proven to cause lower toxicity compared to bare BP. The toxicity mechanism of BP at a cellular level is still unknown, even though the in vivo biological effects of BP are well reported. For instance, a study explored the in vitro and in vivo toxicity of BPNS and the results of the in vitro study on a human bronchial epithelial cell line showed that BPNS interferes with membrane potential, leading to elevated ROS generation which activates caspase-3 leading to cell apoptosis. The in vivo study showed that a single intravenous injection of BPNS does not cause organ damage in mice; however, multiple injections of BPNS led to adverse effects on the mice liver and renal functions [176]. Expertise on development of in vivo animal models will therefore ease in establishing the biocompatibility of BP-related nanomaterials, which in turn will help to speed up the clinical approval.

## 8. Future Prospective

BP, as a vital addition of the family of 2D nanomaterials, has the capability to perform well beyond the (opto) electronics in a range of applications. To improve the stability of BP, the surface outer coating and surface alteration by chemicals and doping have successfully been achieved. Their applications must also be unveiled in the near future in the area of nanoparticles conjugated for targeted gene therapy using aptamers other than miRNA or siRNA, as these area of research have been less explored and have great potential in terms of biomedical applications [151,152]. In the area of advancing medicine, BP-based nanomaterials were also being examined. To date, the cell response to the BP substrate has not yet been reported in terms of the functional properties of cells. The effect of BP film on cell viability, cell differentiation and proliferation would therefore be interesting to study to give a clear interpretation of BP cellular interconnection for regenerative medical applications. Since BP has high electrical properties in comparison with other 2D materials, BP can effectively explore and enhance the electrical conductivity of neurons or cardiomyocytes, by integrating them with BP to enable cardiomyocyte or neural regeneration. In extracellular rigidity, artificial scaffolds such as hydrogels of gelatin methacryloyl (GelMA) with tunable strengths play a significant part in conveying exact mechanical indications to the adherent cells [177,178]. In comparison to those grown on the GelMA polymer, cardiomyocytes inscribed on carbon nanotubes, hybrid hydrogels of graphene and GO-polymers, significantly enhanced morphogenesis of cardiac tissue, caused stronger beating behavior and a greater voluntary beating rate [179,180]. The prospective use of a hybrid hydrogel BP-GelMA scaffold for neural or cardiac tissue regeneration is, therefore, promising [169].

Molecular dynamics (MDs) simulations is one of the specific techniques currently existing to model the weak dispersive hydrogen bonding and polar interactions that control liquid-phase exfoliation (LPE)-based length and time scales of solvent-nanomaterial interfaces, combining unprecedented speed and accuracy with electronic structure simulations of continuum approaches [181]. Quantum density functional theory (DFT) is unable to describe the interaction of electrons well enough to reliably describe the weak van der Waals force to a maximum of a few kJ.mol^−1^. Dispersion modifications can be used to boost this limitation, but somehow the degree of empiricism also correlated with van der Waals forces and remains equivalent to the classical force fields [182,183]. Furthermore, the time scales and size which can be expressed with DFT are much less in the current scenario. Following a theory of colloidal aggregation, Shih et al., in their study, utilized MD simulations to predict the colloidal stability of graphene based on its molecular structure [184]. Kamath and Baker used a particular kind of in silico technique to inspect the efficacy of ionic graphene liquids [185,186,187]. Graphene MD simulation studies reveal various key facts, and similar studies are also required to explore different aspects of BP. All these findings have demonstrated the impact of molecular confinement mostly on kinetics and energetics of aggregation and exfoliation, which highlights the need to consider the solubility energetics in addition to the consequence of solvent-phase microscopic structures. Studies of MD simulations provide insight into different topics, which are important for future development. Other 2D materials other than quantum dots recently showed excellent photothermal conversion efficiencies in cancer therapy. Such 2D materials are found to promote excellent physicochemical properties for different applications, taking into account the resemblances in the function and morphological features between BP and other layered 2D materials of Group III (borophene), Group IV (stanene, germanene and silicene), and Group VA (such as arsene, bismuthene and antimonene) [188]. BP for gas sensing has been demonstrated in various studies as a highly precise material, but its implementation in medical diagnosis is still not investigated thoroughly yet. Hence, upcoming research may also include BP as breath gas analyzer to measure of the levels of nitric oxide and ammonia present in the breath, which allows to detect different diseases such as asthma and renal disease [189]. It would also be fascinating to explore the impact of BP combined with 2D materials in order to construct a combination therapeutic system for cancer therapy. We assume that there would be more research that focuses on use of BP from a biomedical perspective in the near future, with several other remaining challenges and opportunities.

## 9. Conclusions

This present paper provides a summary of BP research for biomedical implementation in the future, including drug delivery, phototherapy, 3D printing, theranostics, bioimaging, and cancer therapy. Layered BP is recognized as a potential candidate to overtake graphene as an innovative and favorable substance for the future nanomedicine and photonics such as photovoltaic cells, photodetectors, and other logical devices, due to its elevated mobility by the carriers, excellent current saturation, controllable bandgap and angular-dependent characteristics. Therefore, in this area, more intense and effective research must be carried out [169]. To date, studies on the implementation of BPs have focused primarily on nanodevices, semiconductors, optoelectronics, and other electronics, whereas significantly fewer attempts have been undertaken to extend its implementation to the other fields, such as biological applications. Regardless of the important progress, studies in this area are still at the budding stage when compared to other well explored 2D material studies. It remains a challenge, from a basic approach, to produce good-quality and bulk BP samples. Hence, for prospective biomedical materials with less toxicity, it is essential to have a tough and efficient method for high value BP bulk production with specific control over its variables, including number of layers, concentration, size, and surface alterations [13].

## Figures and Tables

**Figure 1 nanomaterials-11-00013-f001:**
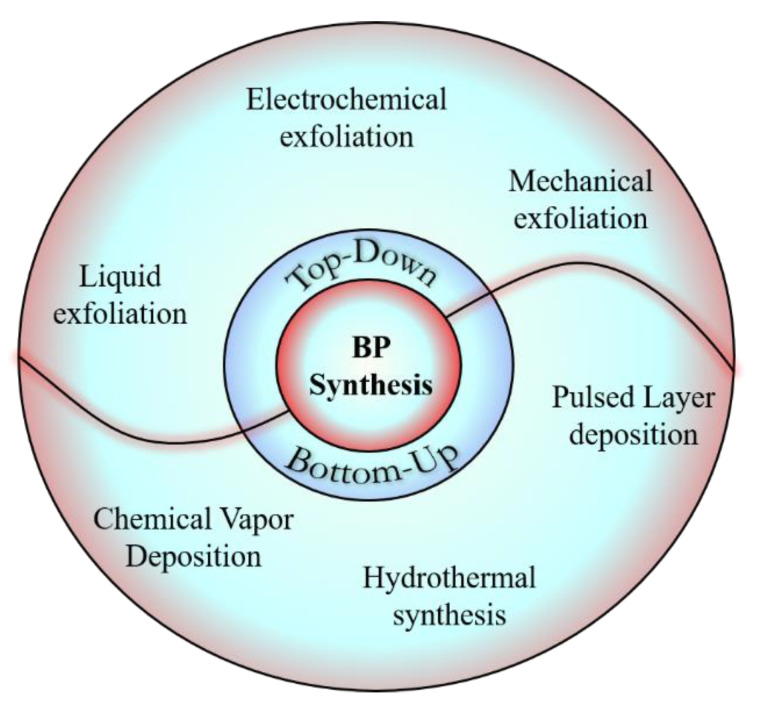
Schematic representation of various synthesis strategies for fabricating two-dimensional (2D) materials.

**Figure 2 nanomaterials-11-00013-f002:**
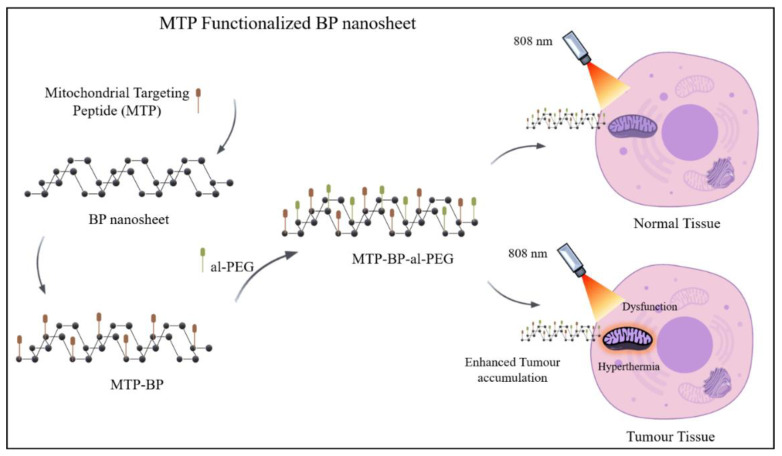
Mitochondrial Targeting Peptide (MTP)-functionalized BP nanosheets for enhanced tumor accumulation and mitochondria targeting.

**Figure 3 nanomaterials-11-00013-f003:**
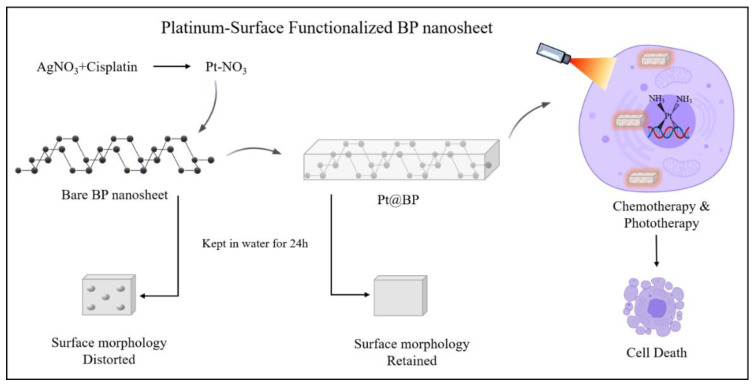
Modified cisplatin surface-modified BP nanosheet for synergistic chemotherapy and phototherapy.

**Figure 4 nanomaterials-11-00013-f004:**
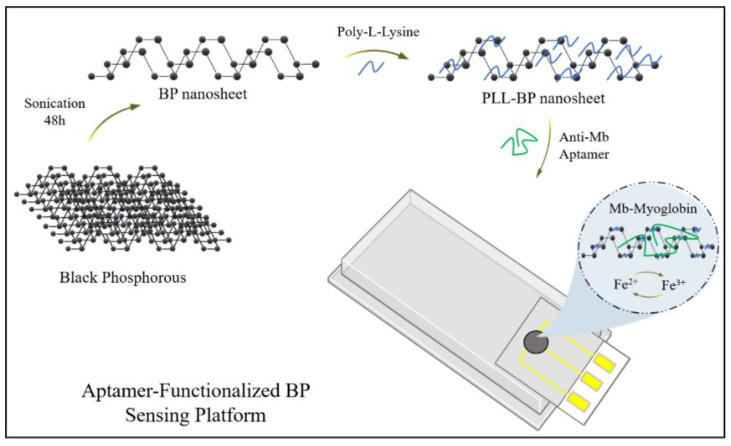
Aptamer-functionalized BP sensing platform for detection of Myoglobin.

**Figure 5 nanomaterials-11-00013-f005:**
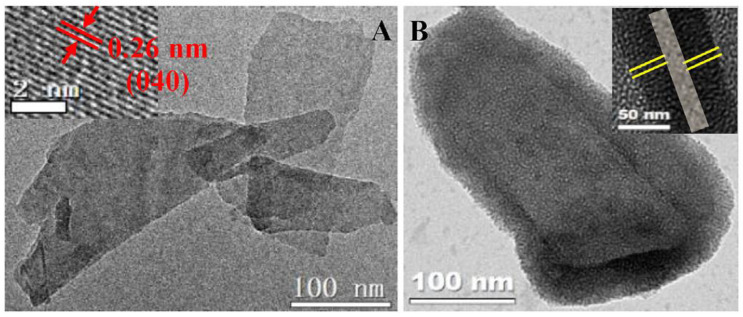
TEM image of (**A**) pristine BP nanosheet, and (**B**) BP@MS (Mesoporous silica). Reproduced with permission from [81]; Copyright Elsevier, 2020.

**Figure 6 nanomaterials-11-00013-f006:**
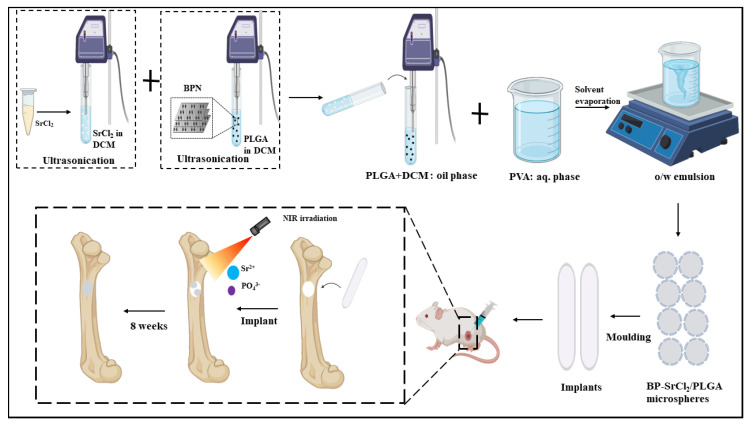
Schematic representation of BP-SrCl_2_/poly D,L-lactic-co-glycolic acid (PLGA) synthesis and its therapeutic application in bone regeneration.

**Figure 7 nanomaterials-11-00013-f007:**
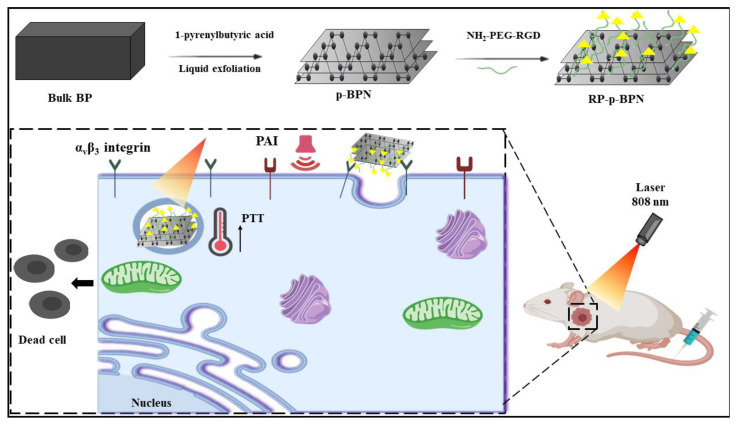
Schematic representation of red phosphorus (RP)-p-Poly-L-lysine (PLL)-coated BP nanosheet (BPNS) synthesis and its therapeutic application in cancer imaging and therapy.

**Figure 8 nanomaterials-11-00013-f008:**
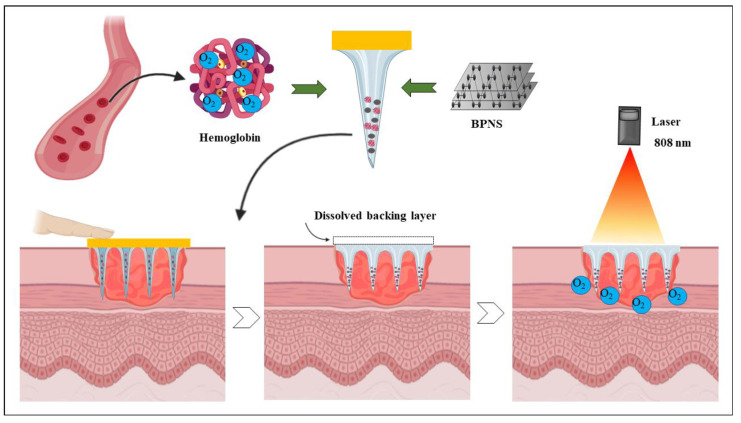
Schematic representation enumerating Near infrared region (NIR)-mediated controlled release of O_2_ for wound healing using microneedles encapsulating BPNS and Hb.

**Figure 9 nanomaterials-11-00013-f009:**
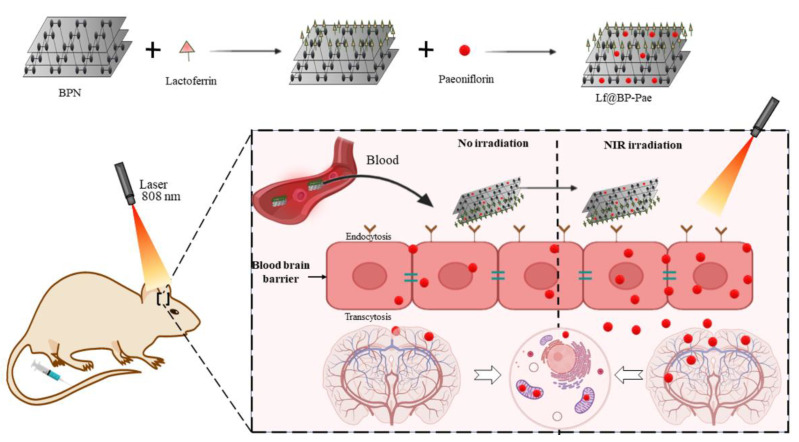
Schematic representation of Lf@BP-Pae synthesis and its therapeutic application in treatment of Parkinson disease.

**Table 1 nanomaterials-11-00013-t001:** Most commonly used methods for the synthesis of black phosphorous (BP) nanomaterials along with their mechanisms, advantages, and disadvantages.

Method	Mechanism	Application Materials	Advantages	Disadvantages	Reference
Liquid exfoliation	Cleavage of multilayered crystals dispersed in a solvent in the presence of ultrasonic energy	Transistors, optical devices, super capacitors, sensors, photocatalysts, bioimaging etc.	Quick, easier, and sturdy to environmental conditions and flexible to scale up.	Difficult to control the thickness of the layered crystals	[22]
Electrochemical Method	Exfoliation of layers occur by applying voltage between anode and cathode that are usually dipped in an acidic solution	Most commonly used for synthesis of graphene from graphite	Various metal 2D nanosheets can be prepared.	Harmful and toxic chemicals are used	[23]
Mechanical Exfoliation Method	A scotch tape process that repeatedly peels off various layers from multilayer crystals/sheets	Imaging, photonic devices, coating, electronic devices, advanced composites, energy storage, metrology, paint, sensors	No chemical is required and eco-friendly	Time consuming, no uniform thickness of the product, low yield, and is not scalable	[24]
Plasma Etching Method	An oxidation process where monolayer is placed on a silica substrate which is exposed to oxygen plasma, due to which the surface layer gets oxidized	Optoelectronics and nanoelectronics	Water soluble, Most commonly used in multiple layered hexagonal carbides and nitrides (MAX) materials	Toxic chemicals such as hydrofluoric acid is used; time consuming	[25]
Chemical Vapour deposition (CVD) Method	A coating process where the precursors, gas, or vapor, can react or decompose on the preselected substrate at high temperature and vacuum in a chamber	For synthesis of 2D nanomaterials and thin films on solid substrates	High-quality product, large surface area	Sensitive and expensive method that involves toxic gases	[26]

**Table 2 nanomaterials-11-00013-t002:** Surface modification of BP and its advantages.

Functionalized Material	Modification Using	Advantages	Reference
BP@FKK Complex	Tripeptide Fmoc-Lys-Lys-Phe (Fmoc-KKF)	improved stability; improved cellular uptake; favorable cell compatibility.	[42]
DFBP nanosheets (MTP-BP-al-PEG)	Mitochondrial Targeting Peptide (MTP) and acid-labile polymer shell	improved stability; ability to accumulate in tumor tissue; target mitochondria.	[43]
Pt@BP	Cisplatin	preserves surface morphology; significant cellular uptake rate; improved cisplatin-resistant cancer cell lines (A2780 and HepG2) drug response.	[44]
PEG-modified BPQDs	PEG	Increased stability in the physiological medium; no observable toxicity.	[45]
FA-PEG@BPQD@DOX	PEG-NH2-FA	Better PTT effect; precise targeting capability for tumor ablation.	[46]
BP@PDA-PEG-FA	HS-PEG-FA; Polydopamine	Synergistic therapy combining chemotherapy with photothermal therapy; enhanced stability; targeting ability for cancer cells.	[47]
PLL-BP-Apt	anti-Mb DNA aptamer	Electrochemical-based sensing for the qualitative and quantitative recognition of the cardiac disease biomarker, Mb; high sensitivity and specificity for Mb; potential in POC diagnosis for cardiac disease management.	[48]
BP@lipid-PEG	Cholesterol	Exhibit broad emissions; sharp contrast and may be utilized to in situ quantify the measurement of blood vessels; potential in photoacoustic (PA) imaging.	[49]

**Table 3 nanomaterials-11-00013-t003:** Summary of applications of BP in therapy of cancer.

Nanocomposite Material	Application	Synthesis of BPNS/BPQD	Cancer type/Cell Lines	Research Outcome	Reference
BPNS	PDT	Liquid-phase exfoliation	MDA-MB-231 cells	BPNS inhibited tumor growth with short exposure of light irradiation due to effective photosensing efficiency	[92]
BP Quantum Dots Resealed erythrocyte nanovesicle (BPQD@RM)	PTT/Immunotherapy	Sonication exfoliation	4T1-breast tumor cells	The combination of PTT with programmed cell death protein delayed the metastatic growth in vivo by increasing activity of CD8+ T cells activity in tumor	[132]
BPNS cellulose hydrogels	PTT	Liquid-phase exfoliation	SMMC-7721 hepatocellular cancer cell lines	The BPNS hydrogel composites displayed excellent PTT efficiency against cancer and was also biocompatible	[133]
PEGylated BPQD	PDT/PTT	Sonication exfoliation	Hep G2 cells	The PEGylated BPQDs showed strong NIR absorption to generate ROS and inhibit tumor thereby showing a theranostic activity in cancer therapy	[118]
BPN/MnO_2_	PDT/PTT/Gene therapy	Modified Liquid-phase exfoliation	HeLa and A549 lung cancer cells	Combinatorial therapy resulted in the release of siRNA to targeted tumor cells by specific nanocomposite degradation and ROS produced by irradiation, led to the suppression of tumor growth.	[134]
PDA@BPQD (Polydopamine functionalized BPQDs)	Photoacoustic imaging (PAI) PAI/PTT	Sonication exfoliation	Nude mice bearing A375 human melanoma tumor	Along with the stability, PDA@BPQDs possessed good photothermal performance on cancer cells	[135]
BPN-CuS-FA (Folic acid anchored CuS nanodot-modified BPN)	PDT/PTT/PAI	Modified mechanical exfoliation	4T1-breast tumor cells	BPN-CuS-FA displayed good PAI activity, enabling in vivo monitoring. The FA targeted folate receptors on the tumor cells enabled the uptake of BPN-CuS complex which showed PDT-PTT-mediated antitumor activity	[136]
PEG@BPN-Ce6 (Chlorin e6 BPN)	PDT/PTT	Liquid-phase exfoliation	HeLa cells	The complex showed good PDT/PTT activity. The in vivo fluorescence imaging displayed that the complex accumulated in tumor cells and synergetic antitumor efficacy was observed due to incorporation of Ce6	[137]
BPNS-bPEI-PEG	PTT/Immunotherapy	Liquid-phase exfoliation	HepG2, RAW264.7 and 4T1 cell lines	The BPNS grafted CpG (immunologic adjuvant) provided as an efficient necroptosis modulator to mediate PTT and anticancer immunotherapy by activating the immunogenic cell death and initiating the immune response.	[138]
NE-BP Neutrophil-coated BP nanoflakes adsorbed on PEI and TGF-β inhibitor	PDT/PTT/Immunotherapy	Liquid-phase exfoliation	4T1 lung cancer cell line	The complex showed enhanced stability, high tumor accumulation, and superior PDT/PTT efficiency in comparison with bare BP resulting in suppressed tumors metastasis.	[139]

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
