# Peer review of "Black Phosphorus as Multifaceted Advanced Material Nanoplatforms for Potential Biomedical Applications"

_nanomaterials, 2020, doi:10.3390/nano11010013_

Round 1

Reviewer 1 Report

The article entitle "Black Phosphorus Nanoplatforms as Versatile Advanced
 Materials for Potential Biomedical Applications: A Focus on  Drug Delivery and Bio-imaging" by Pandey  supposed to deal with application of black phosphorus in drug delivery and imaging. But according to me it does not. This text contains a broad introduction to synthesis of BP, and characterization method and describes functionalization of BP. Only two short part deal with main topic. Thus. this article is not well organized and the title is misleading. Even though there are not so many report in the field of drug deliver authors should make it comprehensive and shorter that reader know only about those specific areas. The author decided to make it longer by adding the extra part that are not so important for the topic and make confusion. Moreover there are already other reviews in the filed

 https://doi.org/10.1002/adfm.201808306

https://doi.org/10.1039/C7MH00305F

10.7150/thno.22573

in the last three years. The authors should focus on the last update in the field and direct the reader to previous reviews. Therefore this review does not bring sth new for the readers and for sure does not assure comprehensive insight in the field of application of BP in the nanomedcine field. The graphical quality of the article is low. The clear description of article content is missing.

Author Response

Reviewer 1

The article entitle "Black Phosphorus Nanoplatforms as Versatile Advanced Materials for Potential Biomedical Applications: A Focus on Drug Delivery and Bio-imaging" by Pandey supposed to deal with application of black phosphorus in drug delivery and imaging. But according to me it does not. This text contains a broad introduction to synthesis of BP, and characterization method and describes functionalization of BP. Only two short part deal with main topic. Thus. this article is not well organized and the title is misleading. Even though there are not so many report in the field of drug deliver authors should make it comprehensive and shorter that reader know only about those specific areas. The author decided to make it longer by adding the extra part that are not so important for the topic and make confusion. Moreover, there are already other reviews in the field. https://doi.org/10.1002/adfm.201808306, https://doi.org/10.1039/C7MH00305F , 10.7150/thno.22573.

In the last three years. The authors should focus on the last update in the field and direct the reader to previous reviews. Therefore, this review does not bring sth new for the readers and for sure does not assure comprehensive insight in the field of application of BP in the nanomedicine field. The graphical quality of the article is low. The clear description of article content is missing

Response to comment 1: The authors are thankful to the reviewer for their time and constructive comments. The authors have re-organized the manuscript along with reshuffling the paragraph to maintain flow. The title of the manuscript has been modified. The latest papers have been cited to keep the manuscript up to date. The synthesis section along with the characterization section has been modified with the latest updates. The images have been modified as per journal guidelines.

Reviewer 2 Report

The submission by Pandey et al. is a comprehensive review of the use of black phosphorus (BP) nanomaterials for various drug delivery and bio-imaging applications. The scope is extensive and covers important subjects such as the synthesis, properties, characterisation, and utilization of BP. Overall the manuscript is well written and organized. References are also up to date. I would therefore like to recommend its publication in Nanomaterials, pending the address of the following comments:

  1. Authors are reminded to cite the relevant references for ALL of the illustrations/ figures that were used in this study.
  2. Section 2 is relatively short compared to the other sections. I would suggest that the author expand on the different sub types of exfoliative techniques (top-down) and wet chemistries (bottom-up) for a more balanced discussion.
  3. Section 4 is supposed to focus on surface modifications of BP. However, the discussion is interlaced with several specific biomedical applications. I suggest the authors to refine section 4 to decouple the modification and application elements. Structure may need to be revised. 
  4. Protein corona formation is a critical determinant that will dictate the bioactivity of nanomaterials. For the benefit of the readers, current state-of-the-art methods to characterize this phenomenon should be added as a sub section in section 5. Addition of such section is also expected to distinguish this review from the other BP-centric reviews. 
  5. There is a disconnect between the title and section 6. The applications discussed in section 6 cover beyond just drug delivery and bio imaging applications…. Authors should revise the title.
  6. Long term in vivo safety and toxic potential of BP is yet another research gap which should be addressed in the future before it can be translated to a clinical settings.

Author Response

The authors have done all the modifications suggested by reviewer

Reviewer 3 Report

The manuscript describes about black phosphorus nanoplatforms as versatile advanced materials for potential biomedical applications with focus on drug delivery and bio-imaging. The manuscript is well structured and detailed however there are certain flaws need to be addressed before publication.

Decision: Minor revision

Comments:

  1. In discussion, authors may describe information of on-going clinical trials if any and commercial aspect of the black phosphorus nanoplatforms.
  2. Authors should also discuss associated challenges with the described technology.

Author Response

The authors have done all the modifications as suggested by reviewer

Round 2

Reviewer 2 Report

The authors have satisfactory addressed the reviewer's comments.